# The Expression of *Triticum aestivum* Cysteine-Rich Receptor-like Protein Kinase Genes during Leaf Rust Fungal Infection

**DOI:** 10.3390/plants12162932

**Published:** 2023-08-14

**Authors:** Ahmed M. Kamel, Khaled Metwally, Mostafa Sabry, Doha A. Albalawi, Zahid K. Abbas, Doaa B. E. Darwish, Salem M. Al-Qahtani, Nadi A. Al-Harbi, Fahad M. Alzuaibr, Hala B. Khalil

**Affiliations:** 1Department of Genetics, Faculty of Agriculture, Ain Shams University, 68 Hadayek Shoubra, Cairo 11241, Egypt; 2Department of Biology, Faculty of Science, University of Tabuk, Tabuk 71491, Saudi Arabiaddarwish@ut.edu.sa (D.B.E.D.);; 3Botany Department, Faculty of Science, Mansoura University, Mansoura 35511, Egypt; 4Biology Department, University College of Tayma, University of Tabuk, P.O. Box 741, Tabuk 47512, Saudi Arabia; 5Department of Biological Sciences, College of Science, King Faisal University, P.O. Box 380, Al-Ahsa 31982, Saudi Arabia

**Keywords:** cysteine-rich receptor-like kinase genes, wheat, leaf rust, resistance, plant–pathogen interactions

## Abstract

Understanding the role of cysteine-rich receptor-like kinases (CRKs) in plant defense mechanisms is crucial for enhancing wheat resistance to leaf rust fungus infection. Here, we identified and verified 164 members of the CRK gene family using the *Triticum aestivum* reference version 2 collected from the international wheat genome sequencing consortium (IWGSC). The proteins exhibited characteristic features of CRKs, including the presence of signal peptides, cysteine-rich/stress antifungal/DUF26 domains, transmembrane domains, and Pkinase domains. Phylogenetic analysis revealed extensive diversification within the wheat CRK gene family, indicating the development of distinct specific functional roles to wheat plants. When studying the expression of the CRK gene family in near-isogenic lines (NILs) carrying *Lr57*- and *Lr14a*-resistant genes, *Puccinia triticina*, the causal agent of leaf rust fungus, triggered temporal gene expression dynamics. The upregulation of specific CRK genes in the resistant interaction indicated their potential role in enhancing wheat resistance to leaf rust, while contrasting gene expression patterns in the susceptible interaction highlighted potential susceptibility associated CRK genes. The study uncovered certain CRK genes that exhibited expression upregulation upon leaf rust infection and the *Lr14a*-resistant gene. The findings suggest that targeting CRKs may present a promising strategy for improving wheat resistance to rust diseases.

## 1. Introduction

Wheat (*Triticum aestivum*) is one of the most important crops all over the world. According to the Food and Agriculture Organization, wheat holds the status of being a crucial crop for over a third of the world’s population, and it contributes to roughly 20% of the protein and calorie intake worldwide. Despite its importance, wheat production faces various challenges, such as diseases, pests, and climate change. To improve wheat yields, researchers are continuously working to generate new resilient cultivars to face diseases and environmental challenges [1].

Wheat rust diseases caused by various fungal pathogens belong to the *Puccinia* genus, leading to severe yield reduction. Among these diseases, leaf rust (*Puccinia triticina*) primarily affects the leaves of wheat plants and has a significant impact if left untreated. Symptoms of leaf rust on wheat plants typically include small, circular, or oval-shaped spots that are orange or reddish-brown [2]. To manage leaf rust disease, it is essential to monitor wheat fields regularly for symptoms of the disease. Wheat breeding for leaf rust disease management is critical for avoiding the fungus. With the continuous emergence of new strains of the fungus that causes leaf rust, the need for improved wheat varieties that are resistant to the disease is more important than ever for developing wheat varieties that are resistant to the disease [3]. The process of selection for resistance can be carried out through conventional breeding or more advanced techniques, such as marker-assisted selection and genetic engineering.

Plants have evolved sophisticated mechanisms to detect and respond to signals by activating immune responses through a set of receptors. Key components of this process are receptor-like protein kinases (RLKs), a large group of receptors that play diverse roles in detecting and integrating extracellular signals and transmitting them to downstream signaling pathways [4]. RLKs perceive various internal and external stimuli and relay the input signal to activate the expression of target genes that are relevant to the detected signals. Typically, an RLK contains at least three protein domains: an extracellular domain that perceives a ligand molecule, a transmembrane domain that anchors the protein in the cell membrane system, and a cellular kinase domain that transmits the signal to downstream regulators primarily through phosphorylation [5].

Cysteine-rich receptor-like protein kinases (CRKs) are a vast subfamily of receptor-like kinases that are commonly known as the domain of unknown function 26 (DUF26) or the stress antifungal domain in RLKs [6]. Their extracellular domains contain two copies of the stress antifungal domain, with a motif sequence of C-X8-C-X2-C that conserves the Cys residues [7]. The presence of two copies of this domain in CRKs suggests a potential for enhanced binding capabilities and signaling interactions with extracellular ligands, such as pathogen-derived molecules.

CRKs have been identified as important contributors to disease resistance and cell death in various plant species [8,9]. In wheat, the CRK genes have been found to be up-regulated in response to different pathogens, such as *Septoria tritici* blotch, *Puccinia triticina*, and *Rhizoctonia cerealis* [10,11,12]. Studies have shown that the overexpression of certain CRK genes in wheat can lead to improved disease resistance. For example, overexpression of the *Ta*CRK3 gene in wheat enhances resistance to *Rhizoctonia cerealis* by promoting the production of reactive oxygen species (ROS), which is important in inhibiting pathogen growth [13]. In addition, *Ta*CRK7 mediates resistance to fusarium crown rot caused by *Fusarium pseudograminearum*. Similarly, overexpression of the *Ta*CRK1 gene in wheat has been shown to enhance resistance to *R. cerealis* by increasing ROS levels and activating the salicylic acid pathway [12]. The role of CRKs in plant defense is not limited to disease resistance mechanisms. CRKs have also been implicated in the regulation of programmed cell death, which plays a crucial role in the plant’s immune response to pathogens [14]. Overall, these findings suggest that CRKs have the potential to serve as useful targets for improving disease resistance in wheat through genetic engineering and breeding programs.

CRKs play a crucial role in protecting wheat against rust diseases. The gene expression study of wheat cultivars infected with rust fungi identified several genes, including CRKs that were significantly upregulated in response to rust infection [15]. For instance, *Ta*CRK2 was induced by both the leaf rust and stripe rust pathogens. The overexpression of *Ta*CRK10 in wheat resulted in enhanced resistance to stripe rust, leading to the activation of several defense-related genes [16]. Another study was also performed on *Puccinia graminis* that detected the overexpression of *Ta*CRK3 in wheat with enhanced resistance. The overexpression of *Ta*CRK3 resulted in increased accumulation of reactive oxygen species (ROS), callose deposition, and lignin biosynthesis, which are all associated with plant defense responses [13]. These studies collectively suggest that the CRK genes play an essential role in wheat defense against rust diseases and that targeting these genes could be a potential strategy for improving rust resistance in wheat.

Near-isogenic lines (NILs) are invaluable tools in plant breeding and genetic research. These lines exhibit genetic variation at a specific locus, typically a resistance gene, while maintaining genetic uniformity across the rest of their genome [17]. This unique characteristic allows researchers to isolate and study the effects of a single gene on various traits and phenotypes [18]. NILs provide a controlled experimental system to investigate the specific gene’s influence, offering insights into the genetic basis of traits and facilitating the development of improved crop varieties. One area where NILs have made significant contributions is in combating wheat leaf rust disease. To combat this disease, plant breeders have utilized NILs to introduce and study resistance genes within the wheat genome. By comparing NILs with and without the specific resistance gene, researchers can elucidate the gene’s impact on disease resistance and better understand the underlying mechanisms of plant defense [19]. This knowledge is then applied to develop resistant wheat varieties through selective breeding, molecular-marker-assisted selection, or genetic engineering techniques, ultimately providing farmers with novel cultivars resistant to leaf rust fungus.

Here, we analyzed and compared the gene expression levels of CRKs in both leaf-rust-fungus-infected and healthy wheat plants. In order to achieve this, we utilized the wheat reference v2.1 from the international wheat genome-sequencing consortium (IWGSC) and identified 164 members of CRKs, which were subsequently assigned to their respective chromosomes. Then, we employed RNA-seq analysis to study the expression patterns of the *Ta*CRK genes. Our findings from this study can help unravel the intricate mechanisms of how the CRK gene family contributes to wheat plant defense against leaf rust fungus infection. This knowledge could ultimately lead to the development of more effective strategies for protecting wheat crops against fungal diseases.

## 2. Results

### 2.1. Genome-Wide Identification and Characterization of the T. aestivum CRK Gene Family

We identified all CRK gene family members in *T. aestivum* from the newly released reference genome v2.1 collected from the IWGSC. A total of 164 putative members were identified as cysteine-rich receptor-like kinase genes, along with 221 alternative forms resulting from splicing. Appendix A represent the 221 CDS sequences of the wheat CRK gene family. All *Ta*CRK proteins were manually verified using conserved domain search, SMART, and InterproScan to define a set of typical CRK proteins that included a signal peptide, at least one cysteine-rich/stress antifungal/DUF26 domain, one transmembrane domain (TM), and one Pkinase domain (Figure 1, Appendix A). Out of the 164 examined members, 150 proteins contained two cysteine-rich/stress antifungal/DUF26 domains, while only 14 proteins were found to have a single DUF26 domain. The proteins encoded by wheat CRK genes comprised 428–754 amino acids, with deduced molecular weights ranging from 47 to 84 kDa, while their isoelectric points (pI) fell between 5.16 and 9.25 (Appendix A). In a prior investigation, the CRK gene family was comprehensively identified in five cereal plants, including wheat reference genome v1.0 [20]. Here, we conducted a comparative analysis between the two released references, which resulted in the identification of four novel members (*Ta*CRK12A, *Ta*CRK26B, *Ta*CRK66D, and *Ta*CRK87D) solely present in RefSeq v2.1 while absent in RefSeq v1.0. On the other hand, eight previously identified CRK members were absent in RefSeq v2.1. The identification of CRK members in RefSeq v2.1 highlighted the importance of keeping genomic databases up to date to accurately reflect the genetic diversity of plant species.

We predicted the subcellular localization of wheat CRK proteins using Deeploc. The analysis revealed that out of the entire set of CRK proteins, approximately 157 proteins were predicted to localize on the cell membrane, indicating that the cell membrane may be the primary site of action for this class of proteins (Appendix A). Meanwhile, five CRKs were predicted to be extracellular, and two proteins were found to be localized in the lysosomes, suggesting an uncanonical role in signaling pathways.

Moreover, the motifs of *Ta*CRK proteins were predicted using MEME. The prediction involved running multiple sequence alignments of the CRK family and searching for conserved motifs to identify specific amino acid residues that are critical to protein function. In the case of wheat CRKs, MEME motif prediction identified ten conserved motifs (Appendix A). The ScanProsite viewer revealed the function of these motifs as serine/threonine protein kinases (motif 1), amidation site (motif 2), protein kinase domain (motif 3), tyrosine kinase phosphorylation site 1 (motif 4), casein kinase II phosphorylation site (motif 5), casein kinase II phosphorylation site (motif 6), casein kinase II phosphorylation site (motif 7), Gnk2-homologous domain (motif 8), N-myristoylation site (motif 9), and cell attachment sequence (motif 10). These conserved motifs provided insight into the function of CRKs for further investigations.

The chromosomal localization of the CRK gene family was also assigned. Here, the distribution of 161 *Ta*CRK genes was not uniform on all chromosomes, with the exception of chromosome 4D, where no such members were detected (Appendix A). While chromosome 2D was observed to have the highest number of members (~17) assigned among all chromosomes (Appendix A, Appendix A). The other three CRK members could not be assigned to any specific chromosome. The gene nomenclature guidelines for *T. aestivum* were adhered when assigning the CRK to homologous genes [21].

### 2.2. Evolutionary Relationship of the T. aestivum CRK Family

Based on the phylogenetic tree topology, we found that the *Ta*CRK gene family has undergone significant diversification, with 11 nested identified clades. The phylogenetic analysis included both *Arabidopsis thaliana* and *Brachypodium distachyon* CRKs to differentiate between *Ta*CRK members. To investigate the evolutionary relationships among *Ta*CRKs, the multiple sequence alignment of *Ta*CRKs, *At*CRKs, and *Bd*CRKs was conducted using ClustalW, and the phylogenetic tree was constructed using the maximum-likelihood method and a JTT-matrix-based model (as described in the method). The distribution pattern of wheat CRKs across the eleven generated clades indicated that this protein family has undergone extensive diversification throughout its evolutionary history (Figure 2). First, a subset of closely related *Ta*CRK and *Bd*CRK members was clustered in subclade 11.1.2. This suggests that these proteins could have similar functions and have evolved from a common ancestor. Notably, subclade 11.2 emerged as a prominent group, encompassing the majority of the *At*CRK proteins. This diversification indicated that these *At*CRKs could evolve unique functional roles that are specific to Arabidopsis plants.

### 2.3. Leveraging NILs for Unveiling the CRK’s Expression by Leaf Rust Infection

The genetic homogeneity of NILs, except for a specific locus (resistant gene), allows for the identification of the impact of specific genes or genomic regions on the expression of the CRK genes involved in the defense response to leaf rust. We employed transcript mapping to count the transcripts belonging to the wheat CRK gene family from publicly available RNA-seq data collected from SRA high-throughput sequencing data that involved leaf-rust-infected and uninfected wheat leaf RNA. After transcript normalization and comparing the abundance of CRK genes for susceptible and resistant interactions, we identified specific members that could play a role in the resistance scenario when wheat is challenged by leaf rust infection. Here, the investigation encompassed various stages of leaf rust infection, starting from the initial infection stage characterized by fungal penetration, hyphae formation, and haustoria formation to the subsequent stage involving sporulation, representing the late phase of infection (Figure 3A). We investigated two SRA public-released RNA-seq libraries to highlight the impact of various *P. triticina* races infecting wheat NILs carrying *Lr57*- and *Lr14a*-resistant genes. In the first dataset, *P. triticina* race 77-5 infected the 7-day-old seedlings of wheat WL711 or WL711-*Lr57* line, and a snapshot of the RNA of infected wheat leaves was collected at various infection stages, including 12 h after infection (hpi) and 1, 2, and 3 days after infection (dai) (Figure 3B, Appendix A). The second dataset compared the late effect of the infection of the avirulent *Pt* isolate 96209 and the virulent *Pt* isolate 95037 on the 20-day-old plants of Tha-*Lr14a* or Tha. line at 8 dai (Figure 3C, Appendix A).

### 2.4. Temporal Gene Expression Dynamics of the CRK Gene Family during Leaf Rust Infection

For understanding the expression profile of the CRK gene family at the germination, penetration, and colonization stages of fungal development on wheat plants, four different time points after infection, 12 hai, as well as 1, 2, and 3 dai, were compared with the preincubation stage (Figure 3A). Across these times, the expression profile of the CRK gene family in wheat cultivar WL711 and WL711 carrying the *Lr57*-resistant gene was compared, as depicted in the histograms in Appendix A. Of 164 CRK genes, only 36 were differentially expressed during these stages of fungal development (Figure 4, Appendix A). A group of genes, namely *Ta*CRK8B2, *Ta*CRK29D, *Ta*CRK6D, *Ta*CRK10D, *Ta*CRK80B, *Ta*CRK05A, *Ta*CRK07B, and *Ta*CRK64D, were upregulated in both susceptible and resistant cultivars upon fungal infection. These genes were found to be upregulated at an early stage (12 hai) and continually increased up to 1 dai in the resistant cultivar, WL711-*Lr57*. After that, the expression of these genes decreased, suggesting their involvement in the plant’s early defense response to the fungal infection, and their induction could serve as an early defense mechanism against the pathogen.

Furthermore, the pairwise correlation analysis conducted among tested libraries demonstrated the significance of the time period ranging from 12 hai to 1 dai in the CRK expression profile, particularly in distinguishing resistant and susceptible wheat cultivars (Appendix A). A remarkably high Pearson correlation value of 0.99 was observed between the 12 hai and 1 dai stages of the infected WL711-*Lr57*. In contrast, for WL711, the expression patterns of CRKs exhibited a substantial switch (correlation value 0.751) between similar stages of infection, suggesting the importance of this stage for fungal resistance.

### 2.5. Lr57 Derived Differential Expression of CRK Genes

*Lr57* differentiated the expression of several wheat CRK genes. The upregulation of *Ta*CRK75D, *Ta*CRK21D, *Ta*CRK81D, *Ta*CRK07D, and *Ta*CRK37B was observed in the resistant interaction compared with the susceptible interaction at 12 hai and 1 and 2 dai, as well as *Ta*CRK79A and *Ta*CRK35D at 2 and 3 dai, respectively (Figure 4). In contrast, the susceptible interaction revealed a contrasting gene expression level at 3 dai. *Ta*CRK46B, *Ta*CRK53D, *Ta*CRK20D, and *Ta*CRK71A exhibited a sudden increase in expression levels compared with the resistant interaction, as well as upregulation of genes such as *Ta*CRK24B at 12 hai, *Ta*CRK88A and *Ta*CRK49B at 1 dai, and *Ta*CRK70A at 2 dai. This intriguing finding indicated the dynamic nature of the plant–pathogen interaction, as well as the intricate balance between susceptibility and resistance mechanisms. The differential expression observed between the resistant and susceptible interactions highlighted the complexity and adaptability of the plant’s immune system during plant–pathogen interactions.

### 2.6. Switching the Response of CRK Genes upon Leaf Rust Infection and in the Presence of Lr14a

Comparative analysis of CRK gene expression profiles in uninfected Tha. and Tha-*Lr14a* revealed similarity, with a total correlation value of 0.89 (Figure 5). On the other hand, leaf rust infection was found to significantly affect the expression profile of CRK genes, particularly in the context of incompatible interactions (correlation: 0.65), where *Lr14a* played a significant role in resistance. Interestingly, it appeared that the recognition of the *Pt* fungal isolate by the *Lr14a*-resistant gene activated distinct pathways that led the CRK genes in the wheat transgenic line to act in a different manner compared with their background Thatcher. This finding was further supported by the presence of similar expression patterns between Tha. and Tha-*Lr14a* when infected with the virulent strain *Pt* isolate 95037 (correlation: 0.91). Both results highlight the influence of both leaf rust infection and the recognition of *Pt* race by the *Lr14a* gene on the CRK gene expression alteration, shedding light on the complex dynamics of plant–pathogen interactions.

### 2.7. The Recognition of the Lr14a Resistance Gene to a Leaf Rust Race Turning on a Set of CRK Genes

The presence of the *Lr14a*-resistant gene switched on a set of 13 CRK genes (*Ta*CRK55D, *Ta*CRK56D, *Ta*CRK19A, *Ta*CRK48B, *Ta*CRK46B, *Ta*CRK53D, *Ta*CRK80D, *Ta*CRK21D, *Ta*CRK40A, *Ta*CRK73B, *Ta*CRK81D, *Ta*CRK35D, and *Ta*CRK81B). These genes were found to be expressed at higher levels in the incompatible interaction of Tha-*Lr14a* with the avirulent *Pt* isolate 96209 as compared with other investigated libraries (Figure 6). These findings indicate the potential involvement of these CRKs in the wheat defense mechanism against *Pt* rust infection, specifically *Pt* isolate 96209, during this stage of infection. Contrary to this, the *Lr14a*-resistant gene exhibited an inability to recognize the virulent strain *Pt* isolate 95037, leading to the absence of activation of previously identified CRK genes. Despite the susceptibility of Tha-*Lr14a* to *Pt* isolate 95037, it exhibited a notable defense response in the host, as evidenced by the upregulation of specific CRK genes, *Ta*CRK11B, *Ta*CRK38A, *Ta*CRK16B, and *Ta*CRK20D, indicating their role in mitigating the severe impact caused by the virulent fungal infection.

### 2.8. Downregulation of a Set of CRK Genes upon Leaf Rust Infection

The fungal leaf rust develops an unknown mechanism to suppress or evade the plant’s defense system, resulting in the downregulation of a set of CRK genes. Here, all infected tested wheat cultivars exhibited a consistent pattern of downregulation of specific CRK genes throughout various developmental stages. In both analyzed datasets, the expression levels of three genes, *Ta*CRK52D, *Ta*CRK42B, and *Ta*CRK02A, were significantly upregulated in uninfected leaves (Figure 4 and Figure 6). In addition, *Ta*CRK02D exhibited a significantly high expression level prior to infection (Figure 4). Notably, these genes displayed high expression levels in both WL711-*Lr57* and its genetic background, WL711, but with a notable bias towards WL711-*Lr57*. Following infection with *Pt* race 77-5, a prominent downregulation of these genes was observed throughout all stages of infection. Similarly, in the RNA-seq experiment involving *Lr14a*, a larger set of *Ta*CRK genes (*Ta*CRK50B, *Ta*CRK65D, *Ta*CRK59A, *Ta*CRK84D, *Ta*CRK17B, *Ta*CRK36D, *Ta*CRK42A, *Ta*CRK4B, *Ta*CRK6A, *Ta*CRK90D, *Ta*CRK44A, and *Ta*CRK83D) exhibited high expression levels under normal conditions in both Tha-*Lr14a* and its background. However, during the infection with both avirulent and virulent *Pt* isolates, these specific genes displayed downregulation of expression (Figure 6).

## 3. Discussion

The investigation of the expression profile of CRK genes in wheat plants during leaf rust fungus infection is a significant step towards understanding their role in plant defense mechanisms. Here, we utilized the wheat reference v2.1 from IWGSC to identify 164 members of the CRK gene family and assign them to their respective chromosomes. To ensure the reliability of the identified CRK proteins, a meticulous verification process was conducted using multiple bioinformatics tools. This verification process confirmed that these proteins possessed the typical characteristics of CRK proteins, including the presence of a signal peptide, at least one cysteine-rich/stress antifungal/DUF26 domain, one transmembrane domain, and one Pkinase domain. The predominance of proteins with two cysteine-rich/stress antifungal/DUF26 domains further supported the existing knowledge that this domain is a defining feature of CRK proteins in plants [6,7,22]. The comparison of the v2.1 and v1.0 references revealed the identification of four novel CRK members solely present in v2.1, emphasizing the importance of keeping genomic databases up to date to accurately represent the genetic diversity of plant species [20]. The subcellular localization predictions using Deeploc unveiled important insights into the primary site of action for CRK proteins. The majority of the proteins were predicted to localize on the cell membrane, aligning with previous studies that suggested their involvement in cell signaling. Additionally, the prediction of extracellular and lysosomal localization for some CRK proteins suggested uncanonical roles [8,9]. The predicted motifs encompassed various domains related to protein kinases, phosphorylation sites, and other functional elements. This knowledge might enable the design of experiments aimed at uncovering the precise molecular functions of CRK proteins.

The chromosomal localization analysis revealed an uneven distribution of *Ta*CRK genes on different chromosomes, except for chromosome 4D, where no members were detected. This observation aligned with previous studies reporting chromosomal rearrangements in wheat genomes, particularly involving chromosome 4D [23]. The loss of *Ta*CRK genes on chromosome 4D might reflect a chromosomal rearrangement event during wheat evolution. Furthermore, chromosome 2D displayed the highest number of *Ta*CRK members assigned among all chromosomes, indicating the influence of differential selection pressures and gene duplication events during wheat evolution.

The evolutionary relationship of the wheat CRK gene family provided insights into the diversification and functional roles of *Ta*CRK proteins throughout wheat’s evolutionary history. Here, a phylogenetic tree was constructed in collaboration with *Arabidopsis* and *Brachypodium* CRKs to evaluate *Ta*CRK members. The multiple sequence alignment of *Ta*CRKs, *At*CRKs, and *Bd*CRKs revealed the distribution pattern of wheat CRKs across various clades, highlighting extensive diversification within the gene family. The presence of 11 nested clades in the phylogenetic tree suggests that *Ta*CRK proteins have undergone significant evolutionary changes, leading to the development of distinct functional roles specific to wheat plants. The clustering of closely related *Ta*CRK and *Bd*CRK members in subclade 11.1.2 indicates their potential shared functions and common ancestry. This observation aligns with previous studies that identified subfamilies of CRKs in Arabidopsis and other plant species, indicating that these proteins may have unique roles in different plant lineages [5]. One study on the evolution of CRKs identified two distinct subfamilies of CRKs in *Arabidopsis*, which are thought to have diverged from a common ancestor before the split of the *Brassicaceae* and *Solanaceae* families [24]. Also, studies have expanded on this work and identified additional CRK subfamilies in other plant species, including rice, Phaseolus, and soybean [25,26,27]. In these studies, the evolutionary history of CRKs in plants was complex and dynamic, influenced by various factors such as gene duplication, gene loss, and functional diversification [28]. The extensive diversification observed in the wheat CRK gene family suggests that these proteins have evolved to fulfill specific functional roles in response to the unique environmental and physiological requirements of wheat plants. Unraveling the molecular mechanisms underlying the functional evolution of CRKs in different plant species, including wheat, remains an important area of future research.

Leaf rust fungal pathogens can severely reduce the yield and quality of wheat, resulting in economic losses and food insecurity. Traditional breeding programs have also been successful in enhancing wheat resistance to rust diseases. For instance, breeders developed rust-resistant wheat varieties through the introgression of resistance genes from wild relatives of wheat [3]. However, genetic engineering opens up new avenues for more precise targeting of specific genes involved in plant immunity to develop crops with enhanced disease resistance [29,30]. Targeting CRKs may present a promising research direction for enhancing wheat resistance to leaf rust diseases. In addition, studying the temporal gene expression dynamics of the CRK gene family during leaf rust development on plants could also provide biomarkers for disease diagnosis and monitoring.

The use of NILs has been instrumental in uncovering the activation of CRK genes in response to leaf rust infection. Here, transcript mapping and gene expression normalization using RNA-seq were used to quantify the abundance of wheat CRKs. By comparing CRK gene expression in susceptible and resistant *Pt–Ta* interactions, CRK genes involved in plant resistance and susceptibility to leaf rust infection could be identified. To highlight the impact of different *Pt* races infecting wheat NILs carrying *Lr57*- and Lr14a-resistant genes, we analyzed two distinct RNA-seq datasets. Our findings revealed the temporal expression dynamics of the CRK gene family during leaf rust infection and development, with a subset of CRK genes that exhibited differential expression during the initial and colonialization stages of infection, suggesting their potential involvement in the plant’s early defense response to the fungal pathogen. This finding agreed with Arabidopsis CRK28 and CRK29, which reflected early responses to defense by flagellin perception [31]. Moreover, we also highlighted the significance of this time period between 12 and 24 hai that is considered for distinguishing the resistant and susceptible interactions. Previously, the time of 1 dai has been identified as a critical stage in the fungal development on wheat plants [32].

The differential expression of CRK genes mediated by *Lr57* demonstrates the impact of this resistant gene on driving the expression of specific CRK genes in response to leaf rust infection. The upregulation of *Ta*CRK75D, *Ta*CRK21D, *Ta*CRK81D, *Ta*CRK07D, and *Ta*CRK37B was consistently observed in the resistant interaction compared with the susceptible interaction, indicating their involvement in the defense response against leaf rust. The *Lr57* gene, originating from *Aegilops geniculata*, provides both seedling resistance and adult plant resistance to leaf rust. It was successfully transferred to chromosome 5D [33]. According to them, *Lr57* has a mysterious role but is known as a pleiotropic gene, conferring resistance to multiple fungal pathogens, including leaf rust. To the best of our knowledge, there is currently no direct evidence linking the *Lr57* gene to CRKs. However, in a study where wheat genes were exposed to *P. triticina*, it was observed that with other genes, CRKs were upregulated in interactions showing resistance [31]. This suggests that CRK genes may play a role in the defense response associated with leaf rust resistance, but more research is needed to fully understand their specific involvement and any potential relationship with *Lr57*.

Studies have provided evidence that the overexpression of CRKs in wheat can significantly boost its resistance to rust diseases. This research highlights the potential of CRKs as important components in enhancing the plant’s defense mechanisms and offers promising insights into improving rust disease resistance in wheat crops. For instance, researchers have introduced the CRK gene *Ta*CRK2 into wheat, resulting in increased resistance to leaf rust and stripe rust [16]. Similarly, the overexpression of *Ta*CRK3 also led to enhanced resistance to stripe rust [13].

Our study provides insights into the dynamic expression switching of CRK genes in response to leaf rust infection and the influence of the *Lr14a* resistance gene. Interestingly, the recognition of the *Pt* fungal isolate by *Lr14a* activated a distinct pathway that led to the initiation of the expression of a set of CRK genes in the wheat transgenic line compared with its background Thatcher. *Lr14a*-mediated resistance during leaf rust infection altered the CRK gene family expression profile (correlation: 0.65, Figure 5). These results highlight the influence of leaf rust infection and the recognition of the race of *Pt* by the *Lr14a* gene on changing CRK gene expression, providing insights into the complex dynamics of plant–pathogen interactions. Furthermore, the presence of the *Lr14a* resistance gene is associated with elevating the expression of a set of CRK genes (*Ta*CRK55D, *Ta*CRK56D, *Ta*CRK19A, *Ta*CRK48B, *Ta*CRK46B, *Ta*CRK53D, *Ta*CRK80D, *Ta*CRK21D, *Ta*CRK40A, *Ta*CRK73B, *Ta*CRK81D, *Ta*CRK35D, and *Ta*CRK81B). These genes exhibited higher expression levels in the incompatible interaction of Tha-*Lr14a* with the avirulent *Pt* isolate 96209 compared with other investigated libraries (Figure 6). These findings suggest the potential involvement of these CRK genes in the wheat defense mechanism against *Pt* rust infection, particularly *Pt* isolate 96209 during the sporulation stage. Conversely, the *Lr14a* resistance gene did not recognize the virulent strain *Pt* isolate 95037, leading to the absence of the activation of previously identified CRK genes. Interestingly, despite the susceptibility of Tha-*Lr14a* to *Pt* isolate 95037, the host exhibited a notable defense response, as indicated by the upregulation of specific CRK genes (*Ta*CRK11B, *Ta*CRK38A, *Ta*CRK16B, and *Ta*CRK20D). This suggests that although Tha-*Lr14a* may not possess direct recognition or resistance to *Pt* isolate 95037, it still activates alternative defense pathways involving these CRK genes to counteract the detrimental effects of the virulent fungal infection. *Lr14a* encodes a protein that is localized to the cell membrane. This protein possesses twelve ankyrin (ANK) repeats and exhibits structural similarities to Ca^2+^ permeable nonselective cation channels [32]. In their investigation, transcriptome analyses demonstrated that *Lr14a* leads to the induction of genes associated with calcium ion binding. This finding suggests the activation of new pathways related to resistance, indicating that *Lr14a* plays a role in triggering and regulating defense mechanisms against pathogens.

Interestingly, fungal leaf rust employs an unknown mechanism to subdue or evade the plant’s defense system, resulting in the downregulation of a group of CRK genes. In infected wheat cultivars, a consistent pattern of downregulation of specific CRK genes was observed across various developmental stages, highlighting the impact of infection on their expression levels. Both datasets revealed the genes *Ta*CRK52D, *Ta*CRK42B, and *Ta*CRK02A, which had significantly higher expression levels in uninfected leaves, but other CRK genes were exclusively downregulated by the type of *Lr*-resistant gene. This suggests that the presence of *P. triticina* can potentially influence the expression of members of the *Ta*CRK gene family, which may play roles in plant development and growth processes. Finally, these observations highlight the intricate relationship between the pathogen and the plant’s molecular machinery involved in development and growth.

Overall, our study explains the expression activation and regulation of CRK genes in response to leaf rust infection, shedding light on the complex interactions between plants and pathogens. By leveraging NILs and transcriptome analysis, we contribute to a better understanding of the molecular mechanisms underlying plant resistance to leaf rust and pave the way for future investigations into the functional roles of CRK genes in plant immunity.

## 4. Materials and Methods

### 4.1. Identification, Domain Architecture, Physicochemical Characterization, Motif Assignment, Chromosomal Localization, and Nomenclature of TaCRKs

The latest released version of the *T. astevium* genome from IWGSC collected from the RefSeq v2.1 assembly (https://urgi.versailles.inrae.fr/download/iwgsc/IWGSC_RefSeq_Assemblies/v2.1/, accessed on 26 July 2023) [34] and annotation v2.1 (https://urgi.versailles.inrae.fr/download/iwgsc/IWGSC_RefSeq_Annotations/v2.1/, accessed on 26 July 2023) were used to search for *Ta*CRK genes. All CRK protein sequences of *A. thaliana* and *B. distachyon* were obtained by querying the TAIR database (http://www.arabidopsis.org/, accessed on 26 July 2023) and the *Pd*GDB database (https://www.plantgdb.org/BdGDB/, accessed on 26 July 2023), respectively. The sequences of CRK proteins collected from *Arabidopsis* and *Brachypodium* were used as queries to perform BLASTP searches against the wheat protein sequence database with a maximum E-value of 1 × 10^−10^ to find all putative *Ta*CRK proteins using Local BLAST [35].

All candidates were manually verified with the conserved domain database to inspect the presence of kinase (PF00069) and cysteine-rich/stress antifungal/DUF26 (PF01657) domains [36]. By eliminating proteins that were not preserved in the latest genome version or, more likely, were considered to belong to another protein family, the *Ta*CRK gene family was collected. Genomic sequences, transcript sequences, and CDS sequences were all obtained. All *Ta*CRK genes were analyzed by the EXPASy online tool (https://web.expasy.org/protparam/, accessed on 26 July 2023) [37] to calculate the number of amino acids, molecular weight, and theoretical isoelectric points (pI). Subcellular localization predictions of the CRK proteins were conducted on Deeploc (https://services.healthtech.dtu.dk/service.php?DeepLoc-2.0, accessed on 26 July 2023 [38], which is a tool designed to predict the subcellular localization of proteins based on their amino acid sequence. The multiple expectation maximization for motif elicitation suite, MEME version 5.5.0 (https://meme-suite.org/meme/, accessed on 26 July 2023) [39], was used to assign the distribution of ten conserved motifs among the CRK proteins. The function of the identified motifs was determined using the ScanProsite viewer [40].

Chromosomal localization of CRK genes in *T. aestivum* of the new and old versions was mapped on respective chromosome sequences obtained from Wheat URGI (https://urgi.versailles.inrae.fr/jbrowseiwgsc/, accessed on 26 July 2023) [41] and represented using MapChart software (https://www.wur.nl/en/show/mapchart.htm, accessed on 26 July 2023) [42]. The CRK genes were named in the order in which they appeared on the chromosomes. The standard recommendation for gene symbolization in *T. aestivum* (https://wheat.pw.usda.gov/ggpages/wgc/98/Intro.htm, accessed on 26 July 2023) was followed while designating homologous genes in wheat.

### 4.2. Multiple Sequence Alignment and Phylogenetic Analysis

The CRK protein sequences of *A. thaliana*, *B. distachyon*, and *T. aestivum* were used for constructing the phylogenetic tree. All protein sequences were aligned by the CLUSTALW program of MEGA11 with 90% conserved sites [43]. The Initial tree(s) for the heuristic search were obtained automatically by applying the neighbor-join and BioNJ algorithms to a matrix of pairwise distances estimated using the JTT model and selecting the topology with the highest log-likelihood value. Then, the phylogenetic tree was constructed using the maximum-likelihood method and a JTT-matrix-based model [44], in which bootstrapping was set to 1000 replicates. The tree is drawn to scale, with branch lengths measured in the number of substitutions per site. All positions containing gaps and missing data were eliminated.

### 4.3. RNA-Seq Data Collection

Publicly available RNA-seq data were collected from the sequence read archive (SRA): the public GenBank repository for high-throughput sequencing data. It contains raw sequence data to access a vast amount of genetic data, allowing for comparative analysis and exploration of genetic variation. From SRA databases, we selected two RNA-seq experiments for analyzing the expression of *Ta*CRK genes under leaf rust infection to ensure that we studied various stages of leaf rust infection. The first dataset involved *T. aestivum* NIL WL711_*Lr57* and its background WL711 infected by *P. triticina* race 77-5 (BioProject: PRJNA328385, data released 10 July 2016 [45]). The second RNA-seq data were for *T. aestivum* NIL Thatcher-*Lr14a* (Tha-*Lr14a*) and its background Thatcher (Tha). Both lines were infected by avirulent and virulent *P. triticina* 96209 and 95037 isolates (BioProject: PRJNA674985, data released 6 November 2020 [46]). Both selected SRA datasets were paired-end Illumina sequences created by the Illumina HiSeq 2000 and NovaSeq 6000 platforms, respectively.

### 4.4. Quality Control

Quality control of all collected data was performed to ensure accuracy and high quality. The quality of the raw sequencing data was assessed using the FastQC toolkit [47]. This step helped to identify low-quality sequences, over-represented sequences, and adapter contamination. The Trimmomatic tool was also employed to trim and filter low-quality sequences and contaminants [48].

### 4.5. Mapping Data

To align the high-quality short reads to the *Ta*CRK reference sequences, a bowtie mapping program was used [49]. The filtered reads were mapped to 221 *Ta*CRK full-length cDNAs, and the gene expression level was presented as TPM (transcript per million reads of library) using StringTie [50]. The TPM values were subjected to differential expression analysis using DeSeq2, a widely used tool for this purpose [51].

### 4.6. Statistical Analysis and Heatmap Generation

The statistical analysis and heatmap figures in this study were generated using the R programming language, version 4.3.0, (https://www.r-project.org/, accessed on 26 July 2023). The CRK genes were considered differentially expressed if they met two criteria: a false discovery rate (FDR) *P*-adjusted value < 0.05 and a log_2_ fold change > log_2_(2). The heatmap figures were created using the heatmap function, which allowed for the visualization of complex patterns and relationships in gene expression using the pheatmap v1.0.12 R package (https://cran.r-project.org/web/packages/pheatmap/index.html, accessed on 26 July 2023 [52]).

## 5. Conclusions

This study significantly contributes to our understanding of the wheat CRK gene family by providing a comprehensive genome-wide identification and characterization. The findings shed light on the properties, subcellular localization patterns, conserved motifs, and chromosomal distribution of *Ta*CRK genes. This knowledge lays the groundwork for further investigations into the molecular mechanisms and biological functions of CRK proteins in wheat and other plant species. The presence of *P. triticina*, the causal agent of wheat leaf rust, has been found to potentially impact the expression of *Ta*CRK genes. Transcript mapping and RNA-seq analysis have allowed for the quantification of wheat CRK gene expression and the identification of CRK genes involved in plant resistance to leaf rust. The differential expression of CRK genes mediated by the *Lr57* and *Lr14a* resistance genes indicates their impact on driving the expression of specific CRK genes in response to leaf rust infection. The presence of the *Lr57* and *Lr14a* resistance genes leads to the activation of a distinct pathway and the upregulation of specific CRK genes, suggesting their involvement in the wheat defense mechanism against leaf rust infection. In contrast, the downregulation or alteration of *Ta*CRK gene expression in response to *P. triticina* infection suggests a potential disruption in normal plant development and growth processes.

## Figures and Tables

**Figure 1 plants-12-02932-f001:**
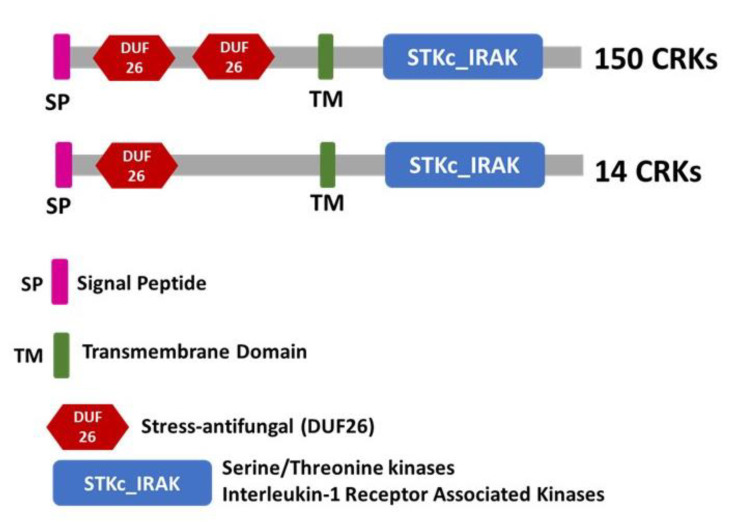
Schematic representation of a typical CRK protein structure. The protein structure consists of a signal peptide (SP) at the N-terminus, followed by at least one cysteine-rich/stress antifungal/DUF26 domain, one transmembrane domain (TM), and one Pkinase domain. A total of 150 proteins contain two cysteine-rich/stress antifungal/DUF26 domains represented by a red hexagonal shape, while only 14 proteins have only one DUF26. The SP is tagged by a pink rectangle; the TM domain is tagged by a green rectangle; and the Pkinase domain is tagged by a blue rectangle.

**Figure 2 plants-12-02932-f002:**
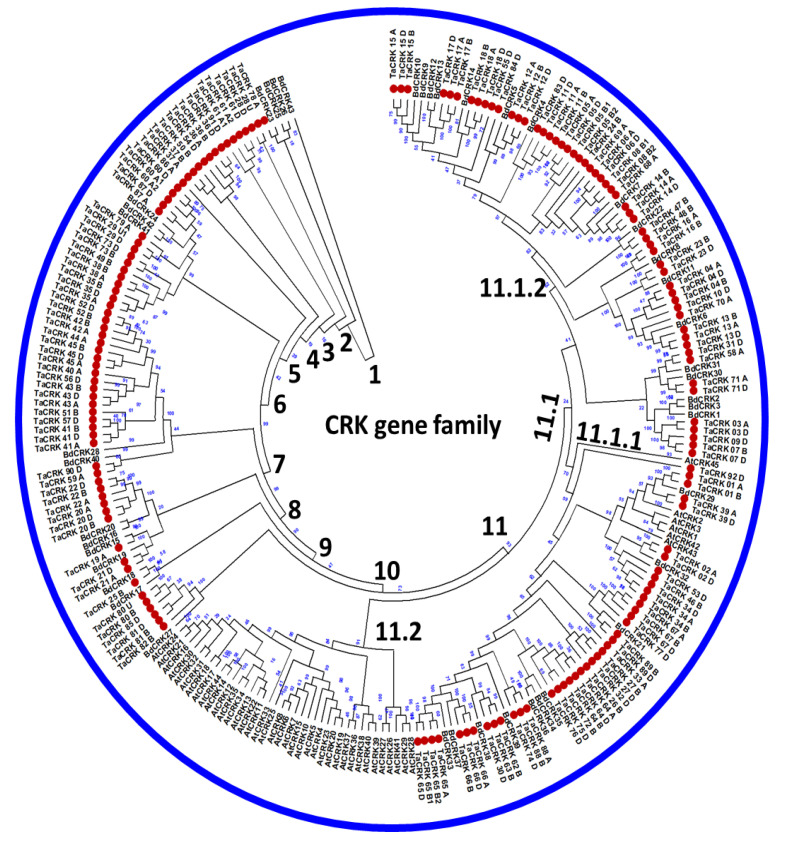
The phylogenetic tree of *T. aestivum* CRK proteins. The tree allowed for the visualization of evolutionary relationships among all *Ta*CRK protein sequences. Protein sequences of *T. aestivum*, *A. thaliana*, and *B. distachyon* were aligned using CLUSTALW (MEGA11) with 90% conserved sites. The maximum-likelihood method and JTT-matrix-based model were used for tree construction (1000 bootstrap replicates). Eleven nested clades of CRKs were generated with extensive diversification. Closely related *Ta*CRK and *Bd*CRK members were clustered in subclade 11.1.2, and a distinct subclade 11.2 separated the majority of *Arabidopsis* CRK proteins from other related species.

**Figure 3 plants-12-02932-f003:**
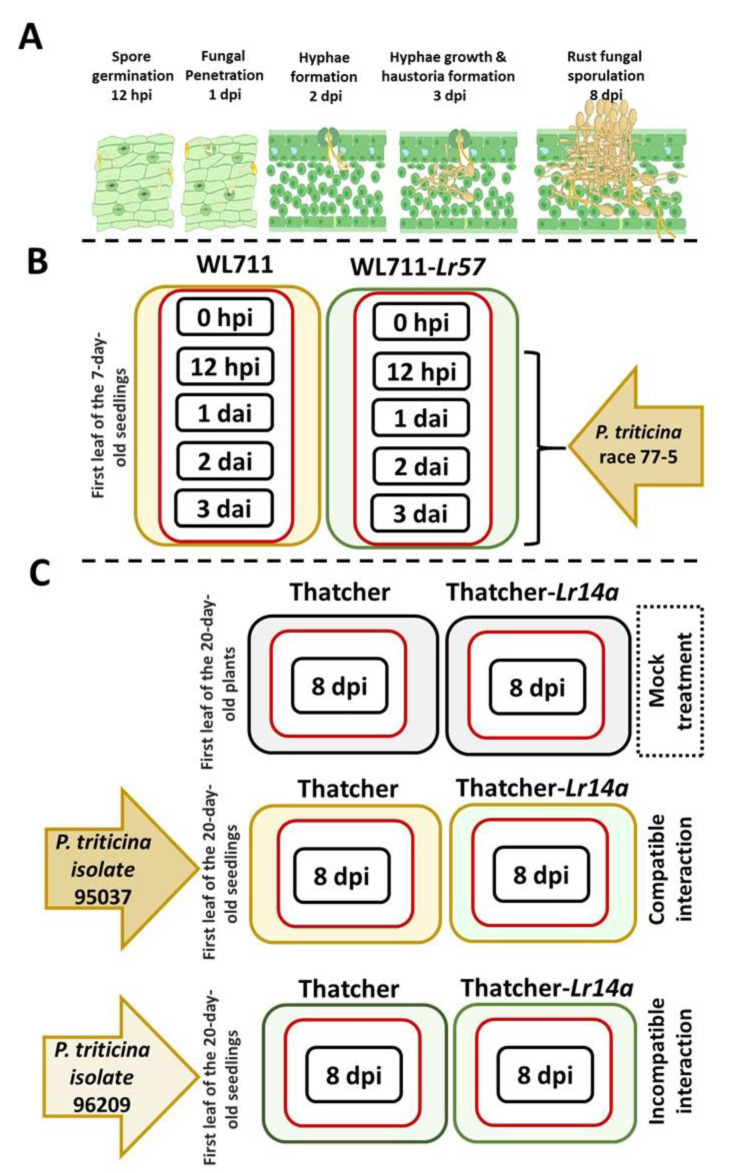
Selected NILs for investigating the response of CRK genes to leaf rust infection: (**A**) Stages of fungal development on wheat leaves. Various phases of leaf rust infection, beginning with the initial stage marked by fungal penetration, the formation of hyphae, and the development of haustoria, then progressing to the sporulation phase. (**B**) The first dataset focused on the *Lr57*-resistant gene. At four leaf rust fungal developmental stages, a snapshot of the leaf RNA present in infected WL711 or WL711-*Lr57* wheat leaves by *Pt* race 77-5 was collected for analysis. (**C**) Wheat Thatcher-*Lr14a* and Thatcher cultivars infected with the avirulent *Pt* isolate 96209 and the virulent *Pt* isolate 95037, representing incompatible and compatible interactions.

**Figure 4 plants-12-02932-f004:**
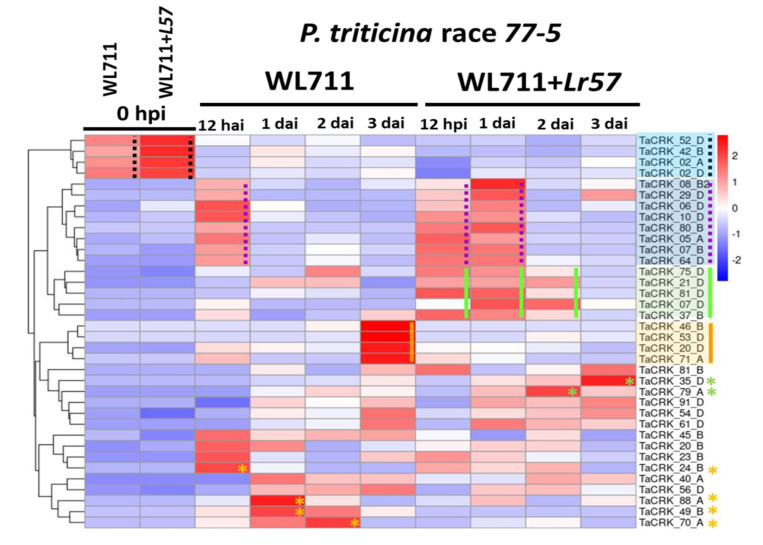
Heatmap of differentially expressed CRK genes during *Pt* race 77-5 development on wheat WL711 and WL711 carrying the *Lr57*-resistant gene. The leaf rust fungal development on wheat leaves at various time points, including 12 hai and 1, 2, and 3 dai. The expression levels of CRK genes were normalized using reads per million (RPM), and the log_2_(2) of RPM values was calculated. The hierarchical clustering expression heatmap was generated in R, where complete linkage was used for clustering. The resulting dendrogram was used to rearrange the rows and columns of the heatmap, placing genes with closely correlated expression profiles next to each other. Red and blue colors represent high and low gene expression levels, respectively. Purple dotted lines indicate genes that are upregulated in both susceptible and resistant cultivars following fungal infection. Green lines and asterisks symbolize genes that are upregulated specifically in the resistant interaction. Orange lines and asterisks symbolize genes that are upregulated in the susceptible interaction. Black dotted lines indicate genes that show elevated expression levels in uninfected leaves.

**Figure 5 plants-12-02932-f005:**
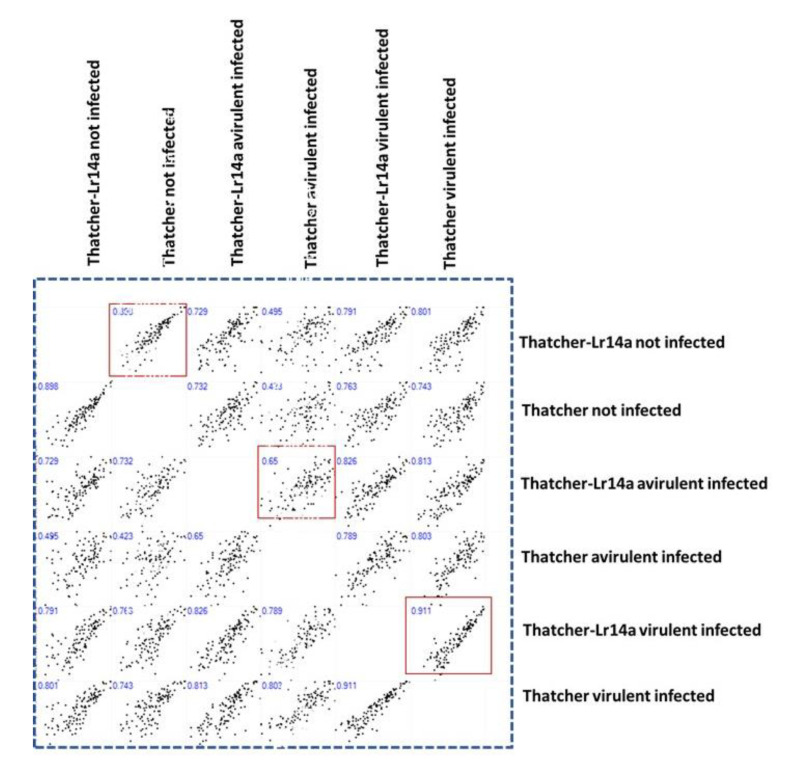
Pearson pairwise correlation of the expression patterns of CRK genes at the late stage of leaf rust fungal infection in wheat Thatcher and Thatcher carrying the *Lr14a*-resistant gene. The avirulent *Pt* isolate 96209 and the virulent *Pt* isolate 95037 infected 20-day-old wheat Tha-*Lr14a* or Tha. lines, representing incompatible and compatible interactions, respectively. The pairwise correlation coefficient of the CRK gene expression pattern between the pairs of tested libraries was calculated and graphed.

**Figure 6 plants-12-02932-f006:**
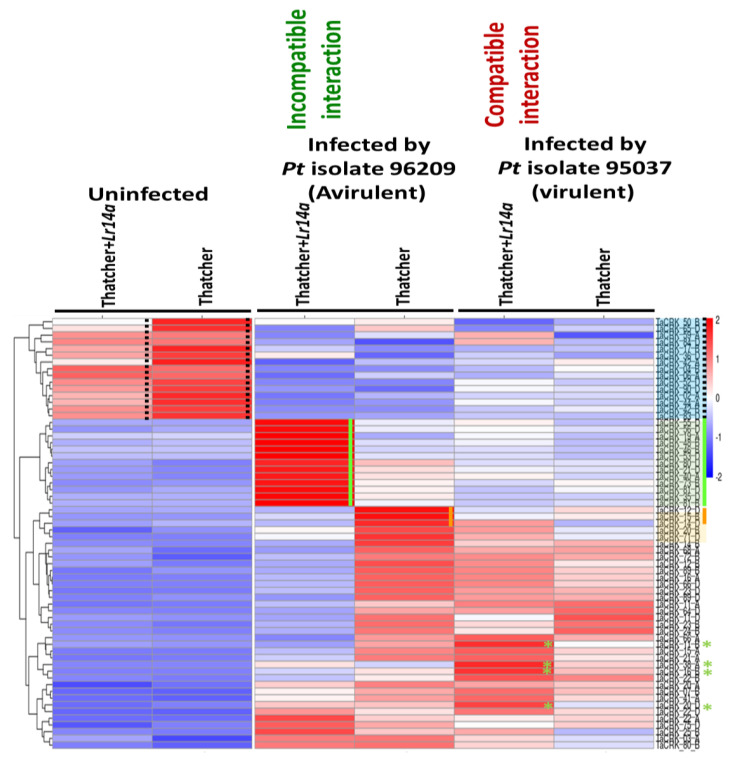
Heatmap of the CRK gene family expression at the late stage of the avirulent *Pt* isolate 96209 and the virulent *Pt* isolate 95037 infections in wheat Thatcher and Thatcher carrying the *Lr14a*-resistant gene. The avirulent *Pt* isolate 96209 and the virulent *Pt* isolate 95037 infected 20-day-old wheat Tha-*Lr14a* or Tha. lines, representing incompatible and compatible interactions, respectively. The expression levels of CRK genes were normalized using reads per million (RPM), and the log_2_(2) of RPM values was calculated. The hierarchical clustering expression heatmap was generated in R, where complete linkage was used for clustering. The resulting dendrogram was used to rearrange the rows and columns of the heatmap, placing genes with closely correlated expression profiles next to each other. Green lines and asterisks symbolize genes that are upregulated specifically in the resistant interaction. Orange lines symbolize genes that are upregulated in the susceptible interaction. Black dotted lines indicate genes that show elevated expression levels in uninfected leaves.

## Data Availability

The data is contained within the manuscript and Appendix A.

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
