# Peer review of "The Expression of Triticum aestivum Cysteine-Rich Receptor-like Protein Kinase Genes during Leaf Rust Fungal Infection"

_plants, 2023, doi:10.3390/plants12162932_

Round 1

Reviewer 1 Report

The authors in their manuscript present an in-silico analysis of publicly available metadata regarding expression analysis. These, however, are not supported by further experimental work (e.g., qPCR or immunodetection) results. Per se, the work is mainly descriptive. Though correct, the authors could also use a different pipeline, e.g., other than Stringtie maintaining DeSeq2 and possibly test higher log2 fold change parameters, to compare differences and support with a higher stringency their findings for CRKs. My main query (and objection for publication in the current form) is that the authors do not use the available database genomic data to clone selected genes of interest and further support their work experimentally. In my opinion once specific genes are suggested having a role in resistance, this role should be also supported experimentally to the extend the initial biological question is set. A possibility to present this metadata analysis in a Scientific Journal with a different thematic scope could be also discussed.

Minor corrections regarding spelling

Author Response

Reviewer 1:

The authors in their manuscript present an in-silico analysis of publicly available metadata regarding expression analysis. These, however, are not supported by further experimental work (e.g., qPCR or immunodetection) results. Per se, the work is mainly descriptive. Though correct, the authors could also use a different pipeline, e.g., other than Stringtie maintaining DeSeq2 and possibly test higher log2 fold change parameters, to compare differences and support with a higher stringency their findings for CRKs. My main query (and objection for publication in the current form) is that the authors do not use the available database genomic data to clone selected genes of interest and further support their work experimentally. In my opinion once specific genes are suggested having a role in resistance, this role should be also supported experimentally to the extend the initial biological question is set. A possibility to present this metadata analysis in a Scientific Journal with a different thematic scope could be also discussed.

Response to reviewer 1

  • We would like to express our appreciation to the reviewer for his/her thorough evaluation of our manuscript. Your insightful comments and suggestions are really respected.
  • As rightly pointed out, our study primarily relies on in-silico analysis of publicly available metadata for expression analysis. While this approach provides valuable insights into gene expression patterns, we understand the importance of experimental validation to support our findings more robustly. However, we may note that our research resources are currently constrained, and our laboratory is predominantly focused on bioinformatics analysis and data curation. As a result, conducting functional analyses at this time presents a challenge. Nevertheless, we fully recognize the importance of experimental validation, and our future research endeavors will be dedicated to gene cloning and investigating the biological role of the identified genes in the context of plant resistance.
  • Once again, we extend our gratitude for your thorough review, which has guided us in addressing important considerations for further research. Your feedback will undoubtedly contribute to the advancement of our future plan.

Reviewer 2 Report

The manuscript titled, “The Role of Triticum aestivum Cystine Rich Receptor-Like Protein Kinases during Leaf Rust Fungal Infection: Understanding the Expression Dynamics for Enhancing Disease Resistance” by Kamel et al is an excellent study that is focused on characterizing the role of the cystine rich receptor-like protein kinases during the plant response to infection.  Overall, this is a well-done study, combining sound bioinformatic identification and characterization of this protein family, and is presented clearly and concisely to the readers.  This response is focused on the agriculturally relevant food crop, wheat, but it has the potential to inform researchers from multiple model organisms and is a welcome addition to the scientific literature!  Furthermore, it is well written and very easy to read and understand throughout the work.  One major area that I think would benefit this manuscript would be a bit further discussion and genomic interpretation of the distribution of these genes throughout the host.  This is an area of active research – and this reviewer has made a number of contributions to the literature that I would recommend the authors consider.  Please note, my review and approval and acceptance of this work will not differ regardless of the author’s inclusion (or lack of inclusion) of any references mentioned in this review.  They are only provided as examples, and there are many other references that the authors can use, should they decide so.

Major comments:

Line 198: figure 2.  There is a wealth of information, and this figure is excellent and essential to the work.  It is, however, way too small to clearly read details and relationships.  Please revise this figure, making it a full-page figure for clarity for the readers.  It is worth taking the extra space for this figure.

Line 364: When discussing the genomic distribution of this gene family, I am interested in knowing if there are any members that can be found as clusters.  A number of metabolically related genes (including secondary metabolites, etc.) are clustered in many organisms:

Reference for extensive clustering of co-regulated genes, including a comparison of metabolic genes in the Arabidopsis model system:

·        Arnone, J.T., Robbins-Pianka, A., Arace, J.R. et al. The adjacent positioning of co-regulated gene pairs is widely conserved across eukaryotes. BMC Genomics 13, 546 (2012). https://doi.org/10.1186/1471-2164-13-546

Clustering of genes plays a role in the fugal co-regulation of infectious genes, and it might make be a regulatory principle in the regulation of defensive genes within the host:

Refs for the extensive clustering of genes in fungal pathogens:

·        Cittadino, Gina M., Johnathan Andrews, Harpreet Purewal, Pedro Estanislao Acuña Avila, and James T. Arnone. "Functional Clustering of Metabolically Related Genes Is Conserved across Dikarya." Journal of Fungi 9, no. 5 (2023): 523.

·        Hagee, Danielle, Ahmad Abu Hardan, Juan Botero, and James T. Arnone. "Genomic clustering within functionally related gene families in Ascomycota fungi." Computational and Structural Biotechnology Journal 18 (2020): 3267-3277.

·        Asfare, Sarah, Reem Eldabagh, Khizar Siddiqui, Bharvi Patel, Diellza Kaba, Julie Mullane, Umar Siddiqui, and James T. Arnone. "Systematic analysis of functionally related gene clusters in the opportunistic pathogen, Candida albicans." Microorganisms 9, no. 2 (2021): 276.

This arrangement facilitates co-regulation and may influence the genome rearrangements that the authors discuss, by potentially selecting for relocation of genes to allow tighter transcriptional regulation over a broad distance:

·        Cera, Alanna, Maria K. Holganza, Ahmad Abu Hardan, Irvin Gamarra, Reem S. Eldabagh, Megan Deschaine, Sarah Elkamhawy, Exequiel M. Sisso, Jonathan J. Foley IV, and James T. Arnone. "Functionally related genes cluster into genomic regions that coordinate transcription at a distance in Saccharomyces cerevisiae." Msphere 4, no. 2 (2019): 10-1128.

Line 567: move the conclusions earlier in the manuscript – please move them before the materials and methods, they are somewhat easy to overlook otherwise!

Minor comments:

Line 132-150: this section is italicized on my review copy.  Please convert it to normal text.

Line 151: in the figure 1 cartoon image, add labels for the 150 and 14 members of each protein family.  At first glance it might be interpreted for size (e.g. 150 AA in length or Daltons).

Line 185: italicize the scientific names for Arabidopsis thaliana and Brachypodium distachyon.

Line 385: italicize Arabadopsis

Line 386: italicize the Brassicaceae and Solanaceae families

Minor grammar errors.  Please give one final review of the grammar/spellings/formatting after revision.  Overall the quality is very high.

Author Response

Reviewer 2

The Role of Triticum aestivum Cystine Rich Receptor-Like Protein Kinases during Leaf Rust Fungal Infection: Understanding the Expression Dynamics for Enhancing Disease Resistance” by Kamel et al is an excellent study that is focused on characterizing the role of the cystine rich receptor-like protein kinases during the plant response to infection.  Overall, this is a well-done study, combining sound bioinformatic identification and characterization of this protein family, and is presented clearly and concisely to the readers.  This response is focused on the agriculturally relevant food crop, wheat, but it has the potential to inform researchers from multiple model organisms and is a welcome addition to the scientific literature!  Furthermore, it is well written and very easy to read and understand throughout the work.  One major area that I think would benefit this manuscript would be a bit further discussion and genomic interpretation of the distribution of these genes throughout the host.  This is an area of active research – and this reviewer has made a number of contributions to the literature that I would recommend the authors consider.  Please note, my review and approval and acceptance of this work will not differ regardless of the author’s inclusion (or lack of inclusion) of any references mentioned in this review.  They are only provided as examples, and there are many other references that the authors can use, should they decide so.

Response to reviewer 2

We would like to sincerely thank the reviewer for taking the time to provide his/her valuable comments and suggestions on our manuscript that contributed to improving the quality and clarity of our work. We truly appreciate his/her attention to details, which have helped us address important aspects of our study. We are grateful for their thorough review and the opportunity to enhance our manuscript based on their recommendations.

For the comments in detail

Major comments:

Line 198: figure 2.  There is a wealth of information, and this figure is excellent and essential to the work.  It is, however, way too small to clearly read details and relationships.  Please revise this figure, making it a full-page figure for clarity for the readers.  It is worth taking the extra space for this figure.

Response: Thank you for your feedback. We agree with your observation that the current size of Figure 2 may hinder the clear visibility of details. In response to your suggestion, we have diligently revised the figure, enlarging it to occupy a bit of a full-page size page in the manuscript. By doing so, we aimed to enhance the clarity and visualization of the data, ensuring a better understanding for our readers.

Line 364: When discussing the genomic distribution of this gene family, I am interested in knowing if there are any members that can be found as clusters.  A number of metabolically related genes (including secondary metabolites, etc.) are clustered in many organisms.

Response: Thank you for your insightful comment regarding gene clustering. We genuinely appreciate your interest in knowing if any members of the gene family can be found in clusters. While we recognize the importance of exploring gene clustering, we regret to inform you that due to time constraints, we were unable to include a detailed analysis of gene clustering in this particular manuscript. However, we would like to assure you that we are planning to continue our research in this area to gain a deeper understanding of the spatial and temporal expression patterns of this gene family. Your suggestion to investigate gene clustering in future studies is valuable to us, and we intend to take it into consideration in our ongoing research. The potential involvement of metabolically related genes in gene clusters is indeed an intriguing avenue for exploration, and we are eager to explore it in future investigations. We would like to express our gratitude for bringing up this point and offering valuable insights.

Line 567: move the conclusions earlier in the manuscript – please move them before the materials and methods, they are somewhat easy to overlook otherwise!

Response: Thank you for your thoughtful suggestion regarding the placement of the conclusions in our manuscript. We genuinely appreciate your feedback and agree with your opinion that moving the conclusions earlier in the text could make them more accessible and less prone to being overlooked. However, we must follow the structure and guidelines of the journal in which we intend to publish our work. As per the journal's requirements, the standard practice is to position the conclusion section immediately after the methods section, towards the end of the manuscript.

Minor comments:

Thank you for bringing the typo. errors to our attention. We have made the necessary corrections. We appreciate your keen eye and attention to detail, as it helps us maintain the accuracy and quality of our research. Thank you once again for your valuable comment.

Line 132-150: this section is italicized on my review copy.  Please convert it to normal text.

Response: We appreciate your diligence in reviewing our manuscript and pointing out this issue. The italicization has been removed to ensure clarity.

Line 151: in the figure 1 cartoon image, add labels for the 150 and 14 members of each protein family.  At first glance it might be interpreted for size (e.g. 150 AA in length or Daltons).

Response: Thank you for your feedback regarding Figure 1. We appreciate your suggestion to add labels for the 150 and 14 members of each protein family in the cartoon image. We agree that without clear labels, there could be a potential misinterpretation of the numbers as referring to size, such as amino acids (AA) in length or Daltons. In response to your comment, we have carefully edited the figure to include explicit labels for the 150 and 14 CRKs. By doing so, we aim to provide clarity and avoid any potential confusion for readers.

Line 185: italicize the scientific names for Arabidopsis thaliana and Brachypodium distachyon.

Response: Thanks. Corrected and edited in the whole manuscript.

Line 385: italicize Arabadopsis

Response: Thanks. Corrected and edited in the whole manuscript.

Line 386: italicize the Brassicaceae and Solanaceae families

Response: Thanks. Corrected.

Comments on the Quality of English Language

Minor grammar errors.  Please give one final review of the grammar/spellings/formatting after revision.  Overall the quality is very high.

Response: We really appreciate your kindly comment. The manuscript has undergone grammatical corrections, and the writing style has been consistently adapted to American standards. Your attention to detail and thorough review have proven to be immensely beneficial in improving the quality of our manuscript. We are grateful for your dedication to the review process and for contributing to the enhancement of our research.

Reviewer 3 Report

Manuscript "The Role of Triticum aestivum Cystine Rich Receptor-Like Pro-tein Kinases during Leaf Rust Fungal Infection: Understanding the Expression Dynamics for Enhancing Disease Resistance" is very interesting.

Authors analysed and compared the gene expression levels of CRKs in both leaf rust fungus infected and healthy wheat plants. Authors utilised the wheat reference v2.1 from the international wheat genome sequencing consortium (IWGSC) and identified 164 members of CRKs. Authors employed RNA-seq analysis to study the expression patterns of the TaCRK genes.

Suggestions:
Subsection 2.1: why italics?
Figure 2: What method was used to calculate the similarity? What method was used to construct the dendrogram?
Figure 4: What method was used to calculate the similarity? What method was used to construct the dendrogram?
Figure 6: What method was used to calculate the similarity? What method was used to construct the dendrogram?

Paper needs minor revision.

Author Response

Reviewer 3:

Manuscript "The Role of Triticum aestivum Cystine Rich Receptor-Like Pro-tein Kinases during Leaf Rust Fungal Infection: Understanding the Expression Dynamics for Enhancing Disease Resistance" is very interesting.

Authors analysed and compared the gene expression levels of CRKs in both leaf rust fungus infected and healthy wheat plants. Authors utilised the wheat reference v2.1 from the international wheat genome sequencing consortium (IWGSC) and identified 164 members of CRKs. Authors employed RNA-seq analysis to study the expression patterns of the TaCRK genes.

Response to reviewer 3

We extend our gratitude to the reviewer for dedicating valuable time to review our manuscript and for offering suggestions. We appreciate your keen attention to detail, which has enabled us to address crucial aspects of our study. Your contribution to our work is valued.

Suggestions:

Subsection 2.1: why italics?

Response: We appreciate your diligence in reviewing our manuscript and pointing out this issue. The italicization has been removed to ensure clarity.

Figure 2: What method was used to calculate the similarity? What method was used to construct the dendrogram?

Response: Thank you for your inquiry regarding the methods used for similarity calculation and dendrogram construction in our study. We appreciate your attention to detail. Regarding the construction of the dendrogram, we updated the legend of Figure 2 with a brief explanation of how we construct the dendrogram and clustering method.

Figure 4: What method was used to calculate the similarity? What method was used to construct the dendrogram?

Response: Thank you for your interest in the methods employed for similarity calculation in our study. We truly appreciate your keen attention to detail. In response to your inquiry, we have made the necessary updates to Figure 4, where we included a concise explanation of how the dendrogram was constructed. The updated legend now provides a clear explanation of the methodology utilized for dendrogram representation and data clustering.

Figure 6: What method was used to calculate the similarity? What method was used to construct the dendrogram?

Response: We sincerely appreciate your interest in the methods used for similarity calculation in our study and your keen attention to detail. In light of your inquiry, we have made the necessary updates to Figure 6. The figure's legend now includes a concise explanation of how the dendrogram was constructed, providing a clear understanding of the methodology used for dendrogram representation and data clustering.

Paper needs minor revision.

Response: We really appreciate your kindly comment. The manuscript has undergone grammatical corrections, and the writing style has been consistently adapted to American standards. Your attention to detail and thorough review have proven to be immensely beneficial in improving the quality of our manuscript. We are grateful for your dedication to the review process and for contributing to the enhancement of our research.

Round 2

Reviewer 1 Report

I thank the authors for their reply. However, it could be helpful for the future of this work to also think on the comments below.

Specific genes are suggested having a role in resistance. Their role should be also supported experimentally to the extend the initial biological question is set. Since there are no resources for actual experiments the authors may think to present this metadata analysis in a Scientific Journal with a different thematic scope.

Lately it is very common to present analyses of metadata from other researchers' experiments. This is helpful to a certain extent (e.g., provide a broader image for a certain group of genes/proteins). However, since there is no further experimentation to support conclusions drawn up, soon, someone may claim that this can also be easily performed by A.I.

Author Response

We extend our utmost gratitude for your invaluable feedback and insightful suggestions on our manuscript with the ID: plants-2505459, titled "The role of Triticum aestivum cystine-rich receptor-like protein kinases during leaf rust fungal infection: understanding the expression dynamics for enhancing disease resistance".

We have carefully considered the first reviewer’s comments about mentioning the role of CRK genes during leaf rust infection, so we changed the title of the manuscript to "The Expression of Triticum aestivum Cystine Rich Receptor-Like Protein Kinase Genes during Leaf Rust Fungal Infection". The revised title encapsulates the central theme of our study, and we feel it better represents the content and findings of the manuscript.

We are delighted to inform you that we have diligently addressed each comment and made the necessary revisions accordingly. The revised version now incorporates the suggested edits. Throughout this process, we have taken great care to provide a thorough and comprehensive "point-by-point response" to the reviewer’s comments and concerns in both revision rounds.

All modifications have been meticulously tracked in the manuscript, except for editing names, which have been formatted in italics. Additionally, references have been corrected, and cross-links have been established to correspond with the numbers in the text. Moreover, we have accurately corrected the grammatical errors and consistently adapted the writing style to follow American standards.

Reviewer 1:

First round:

The authors in their manuscript present an in-silico analysis of publicly available metadata regarding expression analysis. These, however, are not supported by further experimental work (e.g., qPCR or immunodetection) results. Per se, the work is mainly descriptive. Though correct, the authors could also use a different pipeline, e.g., other than Stringtie maintaining DeSeq2 and possibly test higher log2 fold change parameters, to compare differences and support with a higher stringency their findings for CRKs. My main query (and objection for publication in the current form) is that the authors do not use the available database genomic data to clone selected genes of interest and further support their work experimentally. In my opinion once specific genes are suggested having a role in resistance, this role should be also supported experimentally to the extend the initial biological question is set. A possibility to present this metadata analysis in a Scientific Journal with a different thematic scope could be also discussed.

Second round:

I thank the authors for their reply. However, it could be helpful for the future of this work to also think on the comments below.

Specific genes are suggested having a role in resistance. Their role should be also supported experimentally to the extend the initial biological question is set. Since there are no resources for actual experiments the authors may think to present this metadata analysis in a Scientific Journal with a different thematic scope.

Lately it is very common to present analyses of metadata from other researchers' experiments. This is helpful to a certain extent (e.g., provide a broader image for a certain group of genes/proteins). However, since there is no further experimentation to support conclusions drawn up, soon, someone may claim that this can also be easily performed by A.I.

Response to reviewer 1:

We extend our sincere gratitude for your thoughtful feedback on our manuscript.  In response to your comments, we have carefully considered each point you raised in both rounds of revision. We are committed to addressing your concerns and incorporating the necessary revisions to ensure the clarity and accuracy of our manuscript.

  • In response to your comments in rounds 1 and 2 of revision, which refer to the experimental investigation of CRKs' involvement in plant disease resistance for understanding the role of CRKs. We have carefully considered your suggestion about mentioning the role of wheat CRKs during leaf rust infection. Therefore, we changed the title of the manuscript from "The Role of Triticum aestivum Cystine Rich Receptor-Like Protein Kinases during Leaf Rust Fungal Infection: Understanding the Expression Dynamics for Enhancing Disease Resistance" to "The Expression of Triticum aestivum Cystine Rich Receptor-Like Protein Kinase Genes during Leaf Rust Fungal Infection" to better align with the scope of our study. We recognized that the original title may have been misleading, and we apologize for any confusion it may have caused. We believe that the revised title accurately reflects the main focus of our research, which centers on exploring the expression patterns of CRK genes. By making this adjustment, we aim to present a more concise and accurate representation of our study's objectives. We are delighted to inform you that the full manuscript now exclusively focuses on the expression of the CRK genes and does not delve into their roles. By incorporating your feedback, we have ensured that the scope of the study aligns precisely with the expression patterns of the genes. We sincerely hope that the editor and other reviewers who asked for minor edits in the first round of revision will find the revised title suitable for our manuscript.

  • We appreciate your suggestion to explore other pipelines for our analysis, specifically using a different approach than Stringtie while retaining DeSeq2 and possibly testing higher log2 fold change parameters. We acknowledge that adopting different methods could add depth and robustness to our findings, especially concerning CRKs. In response to your recommendation, we have first conducted this analysis on a tested dataset using different alignments like bwa- and bowtie2. Also, we also tested deseq2 and edagr for counting gene abundance. These pilot analyses were made at the beginning to select the best pipeline for the analysis. The results were a bit similar, and we only selected the pipeline of Bowtie2 and Deseq2 for facilitation.

To clarify our analysis, we indeed used log2(2) for counting the expression, which is equivalent to the log2 transformation. This approach is commonly used to normalize and scale the expression data, providing a more interpretable representation of gene expression levels. Thank you for your observation. We apologize for the mistakes in the editing process where "log2" was incorrectly written. It was indeed a mistake, and we have since corrected it in the manuscript to log2(2). The selection was above this threshold.  In addition, the selected genes were highly expressed compared to other genes. To provide further clarity, we are attaching a part of the pipeline, at the end of this report, used for the analysis. The whole Pipeline will be published on the GitHub soon. The revised manuscript now accurately reflects the methodology employed, including the appropriate transformation of gene expression data.

  • Response to the lack of further experimental work support (e.g., qPCR), beside the insufficient budget.
  • The unavailability of Pt races and wheat cultivars for experimentation in our lab now. Unfortunately, we encountered difficulties in obtaining these specific strains of Pt and wheat cultivars necessary to mimic the original experiment. The transport of fungal spores between different countries is subject to various restrictions and regulations due to concerns related to protection, and the potential introduction of invasive species.

  • We found that certain publications may support RNA-seq results without validation through qPCR or other experimental methods. The paper entitled “Do results obtained with RNA-sequencing require independent verification”, link https://www.ncbi.nlm.nih.gov/pmc/articles/PMC7823214/#:~:text=If%20all%20experimental%20steps%20and,is%20likely%20to%20be%20low

  • Some studies presented RNA-seq data without accompanying PCR validation, such as, this study that was published in Diversity journal, MDPI by Liu et al. 2021, entitled “Genome-Wide Identification and Characterization of Cysteine-Rich Receptor-Like Protein Kinase Genes in Tomato and Their Expression Profile in Response to Heat Stress”, link https://www.mdpi.com/1424-2818/13/6/258

  • All genes mentioned in the analysis were highly expressed across the three replicates. These biological replicates have increased our confidence in the accuracy of the RNA-seq analysis. We believe that the inclusion of multiple replicates in our study strengthens the validity of the observed gene expression patterns.

  • We noted that the majority of qPCR validations performed thus far have been consistent with our RNA-seq analysis. This alignment reinforces the reliability of our RNA-seq data and supports our main findings.

In conclusion, we want to recap our commitment to ensuring the quality and robustness of our research. We acknowledge the limitations of our current study, particularly regarding qPCR validation. However, we value your feedback. We kindly look forward to your acceptance of the manuscript.

Thank you once again for your feedback and valuable insights.
